



# The DOE E3SM Version 2.1: Overview and Assessment of the Impacts of Parameterized Ocean Submesoscales

Katherine M. Smith[1], Alice M. Barthel[1], LeAnn M. Conlon[1], Luke P. Van Roekel[1], Anthony Bartoletti[2], Jean-Christophe Golaz[2], Chengzhu Zhang[2], Carolyn Branecky Begeman[1], James J. Benedict[1], Gautam Bisht[3], Yan Feng[4], Walter Hannah[2], Bryce E. Harrop[3], Nicole Jeffery[1], Wuyin Lin[5], Po-Lun Ma[3], Mathew E. Maltrud[1], Mark R. Petersen[1], Balwinder Singh[3], Qi Tang[2], Teklu Tesfa[3], Jonathan D. Wolfe[1], Shaocheng Xie[2], Xue Zheng[2], Karthik Balaguru[3], Oluwayemi Garuba[3], Peter Gleckler[2], Aixue Hu[6], Jiwoo Lee[2], Ben Moore-Maley[1], and Ana C. Ordoñez[2]

[1]Los Alamos National Laboratory, Los Alamos, NM, USA
[2]Lawrence Livermore National Laboratory, Livermore, CA, USA
[3]Pacific Northwest National Laboratory, Richland, WA, USA
[4]Argonne National Laboratory, Lemont, IL, USA
[5]Berkeley Livermore National Laboratory, Berkeley, CA, USA
[6]National Center for Atmospheric Research, Boulder, CO, USA

**Correspondence:** Katherine M. Smith (kmsmith@lanl.gov)

**Abstract.** The U.S. Department of Energy's Energy Exascale Earth System Model (E3SM) version 2.1 builds on E3SMv2 with several changes, the most notable being the addition of the Fox-Kemper et al. (2011) mixed layer eddy parameterization. This parameterization captures the effect of finite-amplitude, mixed layer eddies as an overturning streamfunction and has the primary function of restratification. Herein, we outline the changes to the mean climate state of E3SM that were introduced by the addition of this parameterization. Overall, the presence of the submesoscale parameterization improves the fidelity of the v2.1 simulation by reducing the North Atlantic ocean surface biases present in v2, as illustrated by changes to the climatological sea surface temperature and salinity, as well as Arctic sea-ice extent. Other impacts include a slight shoaling of the mixed layer depths in the North Atlantic, as well as a small improvement to the Atlantic Meridional Overturning Circulation (AMOC). We note that the expected shoaling due to the parameterization is regionally dependent in our coupled configuration. In addition, we investigate why the parameterization and its impacts on mixed layer depth have little impact on the simulated AMOC: despite increased dense water formation in the Norwegian Sea, only a small fraction of the water formed makes its way south into the North Atlantic basin. Version 2.1 also exhibits small improvements in the atmospheric climatology, with smaller biases in many notable quantities and modes of variability.

## 1 Introduction

The U.S. Department of Energy (DOE) Energy Exascale Earth System Model (E3SM) aims to meet the energy mission and science needs of the DOE using state-of-the-art DOE computing resources. Version 1 (E3SMv1) was released in 2018 and, while the land model and coupler were similar to those in CESM2 (Community Earth System Model, Hurrell et al. (2013); Dan-





abasoglu et al. (2020)), the river routing, ocean, sea ice, atmospheric physics, atmospheric dynamical core, and stratospheric chemistry were significantly different. Both lower (110 km atmosphere and 60-30 km ocean) and higher (25 km atmosphere and 18-6 km ocean) resolution configurations were released (Golaz et al., 2019; Caldwell et al., 2019), along with biochemical and cryosphere configurations (Burrows et al., 2020; Comeau et al., 2022). Following version 1, version 2 (E3SMv2) was released in 2022 with significant improvements to the modeled climate, including a 2x speedup from E3SMv1 (Golaz et al., 2022). For this version, a lower resolution configuration and a North American regionally refined configuration (Tang et al., 2023) have been released, with plans for a biogeochemistry configuration with interactive carbon and nutrient cycles, and a cryosphere configuration with regional refinement over the Southern Ocean in the future.

Version 2.1 (E3SMv2.1) builds on E3SMv2 (Golaz et al., 2022) with several changes, most notably the addition of the so-called "Fox-Kemper2011" mixed layer eddy (MLE) parameterization (hereafter referred to as FK11; Fox-Kemper et al. (2008); Fox-Kemper et al. (2011)). Shallow, ageostrophic baroclinic instabilities, often referred to as submesoscale instabilities, develop on lateral density fronts in the weakly stratified surface mixed layer. Once they become finite amplitude, the resulting mixed-layer eddies slump the fronts, releasing potential energy and contributing to the restratification and shoaling of the mixed layer (Boccaletti et al., 2007). Due to their small spatial scales ($\mathcal{O}(10\text{km})$), these submesoscale instabilities and their effects are not explicitly resolved in global ocean models, even at "eddy-resolving" resolutions, and thus need to be parameterized. Fox-Kemper et al. (2008) proposed a parameterization in the form of an overturning streamfunction to mimic the MLE fluxes of density and other tracers. By construction, this overturning streamfunction acts to slump isopycnals and enhance restratification of the mixed layer. This parameterization has been implemented in several other global ocean general circulation models, such as the Parallel Ocean Program (POP) model (Smith et al., 2010), the Modular Ocean Model (MOM) (Griffies, 2009; Adcroft et al., 2019), the Generalized Ocean Layered (GOLD) model (Adcroft and Hallberg, 2006), and the MIT General Circulation Model (MITgcm) (Marshall et al., 1997). According to Fox-Kemper et al. (2011), the general impacts of the parameterization within these models are the shoaling of the mixed layer (with greatest effects in polar winter regions), a strengthening of the Atlantic Meridional Overturning Circulation (AMOC), a reduction in tracer ventilation, small changes to sea surface temperature (SST) and air-sea fluxes, and a reduction in sea-ice basal melting.

In this paper, we largely focus on documenting the implementation of the MLE parameterization from FK11 in the ocean component of the E3SM, the Model for Prediction Across Scales - Ocean (MPAS-Ocean). We investigate the response of the coupled model to the MLE fluxes, with a particular focus on high-latitude convection and large-scale ocean circulation, including the AMOC.

## 2 Methods

E3SMv2.1 implemented several changes from v2, including several bug fixes and additional options that are detailed in Appendix B. However, the primary notable difference from E3SMv2 is the inclusion of the FK11 MLE parameterization, outlined below (Fox-Kemper et al., 2011).


## 2.1 Mixed Layer Eddy Parameterization

The FK11 MLE parameterization captures finite-amplitude, mixed layer eddies as an overturning streamfunction and has the primary function of restratification. It applies a submesoscale transport velocity through a streamfunction given as

$$\Psi = C_e \frac{\Delta_S}{L_f} \frac{H^2 \nabla \overline{b}^z \times \hat{\mathbf{z}}}{\sqrt{f^2 + \tau^{-2}}} \mu(z), \tag{1}$$

$$\mu(z) = \max\left\{0, \left[1 - \left(\frac{2z}{H} + 1\right)^2\right]\left[1 + \frac{5}{21}\left(\frac{2z}{H} + 1\right)^2\right]\right\}, \tag{2}$$

where $C_e$ is an efficiency coefficient, $\Delta_S$ is the local model grid-scale dimension, $L_f$ is an estimate of the typical local width of mixed layer fronts, $H$ is the mixed layer depth, $\overline{b}^z$ is the depth-average buoyancy over the mixed layer, $\hat{\mathbf{z}}$ is the unit vertical vector, $f$ is the Coriolis parameter, and $\tau$ is the time needed to mix momentum across the mixed layer. While recent work has been done to dynamically predict $L_f$ (Bodner et al., 2023), we use a constant value in time and space here in this study.

## 2.2 Model Setup

In order to test the impact of the above changes, we will compare simulations from the E3SMv2 (Golaz et al., 2022) and E3SMv2.1 configurations. As with E3SMv1 and v2, we will focus on climate metrics within the standard resolution configuration, which consists of a 110 km atmosphere, a 165 km land, 0.5° river routing model, and a variable resolution ocean and sea ice mesh going from 60 km in the mid-latitudes to 30 km at the equator and poles. Similarly, the vertical grid remains the same as E3SMv1 and v2, with 72 layers in the atmosphere (∼60 km top) and 60 layers in the ocean (10 m near-surface resolution). Just like v2, the ocean model uses the Gent-McWilliams (GM; Gent and Mcwilliams, 1990) and Redi parameterizations (Redi, 1982), to simulate mesoscale eddies, in addition to the MLE parameterization outlined above.

## 2.3 Simulations

Table 1 summarizes the v2 and v2.1 simulations used in this paper. The full Diagnosis, Evaluation and Characterization of Klima (DECK; Eyring et al. (2016)), five historical simulations, and two idealized $CO_2$ simulations were run for each configuration. In addition, for the purpose of this paper and investigation into the mechanisms of change due to the FK11 MLE parameterization, we ran an additional 33-year extension on both the v2 and v2.1 *piControl* simulations that included several passive tracers (details provided below in Section 4). Historical ensemble members (*v2/v2.1 historical_N*) and the extension runs (*v2/v2.1 piControl Ext*) were initialized from the *v2/v2.1 piControl* on Jan 1 of the year indicated in Table 1. Both the v2 and v2.1 simulations were completed on the DOE-E3SM owned "Chrysalis" machine, which is a traditional high performance computer with 528 nodes with two AMD Epyc 7532 processors per node (64 cores per node) located at Argonne National Laboratory. Computational performance between the two configurations is comparable, with v2 and v2.1 producing 42 simulated years per day and 40.5 simulated years per day on average, respectively, on 105 nodes of Chrysalis.





**Table 1.** Summary of E3SMv2 and E3SMv2.1 Simulations

| Label | Description | Period | Ens. | Initialization |
|---|---|---|---|---|
| *v2 piControl* | Pre-Industrial control with v2 code base | 500 yr | - | Pre-industrial v2 spin-up |
| *v2 historical_N* | Historical period with v2 code base | 1850-2014 | 5 | *v2 piControl* (yrs 101, 151, 201, 251, 301) |
| *v2 piControl Ext* | 33-year piControl extension with v2 code base | 501-533 | - | *v2 piControl* (yr 500) |
| *v2 1pctCO2* | Prescribed 1% $yr^{-1}$ $CO_2$ increase | 150 yr | 1 | *v2 piControl* (101) |
| *v2 abrupt-4xCO2* | Abrupt $CO_2$ quadrupling | 150 yr | 1 | *v2 piControl* (101) |
| *v2.1 piControl* | Pre-Industrial control with v2.1 code base | 500 yr | - | Pre-industrial v2.1 spin-up |
| *v2.1 historical_N* | Historical period with v2.1 code base | 1850-2014 | 5 | *v2.1 piControl* (yrs 101, 151, 201, 251, 301) |
| *v2.1 piControl Ext* | 33-year piControl extension with v2.1 code base | 501-533 | - | *v2.1 piControl* (yr 500) |
| *v2.1 1pctCO2* | Prescribed 1% $yr^{-1}$ $CO_2$ increase | 150 yr | 1 | *v2.1 piControl* (101) |
| *v2.1 abrupt-4xCO2* | Abrupt $CO_2$ quadrupling | 150 yr | 1 | *v2.1 piControl* (101) |





## 3 Overview of v2.1 Climate

In this section, we examine the climatological state of the v2.1 configuration. In particular, we focus on changes to the mean
climate state that were introduced by the addition of the ocean MLE parameterization on the North Atlantic. For that purpose,
we highlight the biases (with respect to observational estimates) in the context of the historical biases that were present in the
v2 configuration.

### 3.1 Oceanic Climate

For a global evaluation of relevant oceanic fields and a comparison between the v2 and v2.1 configurations, we first present a
spatio-temporal root mean squared error (RMSE) in Figure 1. Annual ocean climatologies and RMSE are constructed using the
five member *v2/v2.1 historical* ensemble means over a 1980-2014 period. Observational data are drawn from merged Hadley
Center-NOAA/OI data from 1870-2008 for the sea-surface temperature (SST) (Hurrell et al., 2008), from the NASA Aquarius
satellite from 2011-2015 for sea-surface salinity (SSS) (Lagerloef et al., 2015), from merged absolute dynamic topography
satellite data provided by AVISO (Archiving, Validation and Interpretation of Satellite Oceanographic data; processed by
SSALTO/DUACS and distributed by AVISO+ (https://www.aviso.altimetry.fr) with support from CNES) from 1992-2010 for
sea-surface height (SSH), from ARGO floats from 2000-2017 for the mixed layer depth (MLD) (Holte and Talley, 2009), and
Global Drifter Program drifters from 1979-2015 for near-surface eddy kinetic energy (EKE) (Laurindo et al., 2017). Although
they are relatively modest, all quantities except for SSH see a small global bias reduction due to the presence of the MLE
parameterization. This reduction is seen across all ensemble members. In order to understand regionally where these bias
changes occur, we next dive into a series of ocean climatological maps.





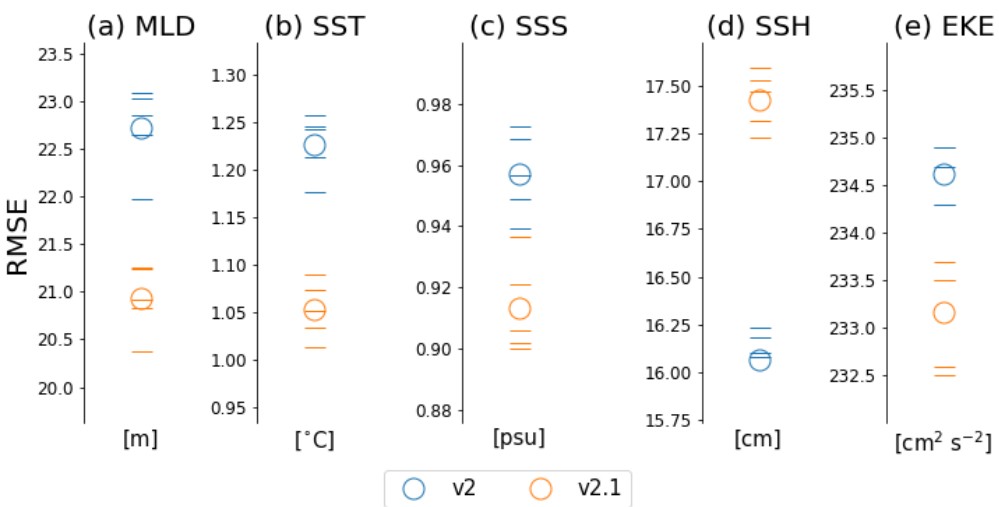

**Figure 1.** RMSE of the global MLD (m), SST (°C), SSS (psu), SSH (cm), and EKE (kg$^2$ s$^{-2}$) for the v2 (blue markers) and v2.1 (orange markers) configurations. Horizontal line markers indicate the metric obtained from individual ensemble members and open circle markers indicate the multi-realization averages of the five *v2/v2.1 historical* ensemble members over a 1980-2014 time period.





In the following, we present a series of ocean climatological maps where panel (a) is the v2 configuration bias in comparison to observations, panel (b) is the v2.1 configuration bias in comparison to observations, and panel (c) is the change in biases between the v2.1 and v2 configurations (ie. positive (negative) values are an increase (decrease) in bias in the v2.1 quantity compared to v2). In these fields, we mask out values considered not statistically significant according to a one-sample T-test (and a two-tailed critical value at alpha = 0.05) when comparing the model ensemble to observations, and a two-sample T-test when comparing the two model ensembles. Since most of the significant oceanic changes from v2 to v2.1 are within the North Atlantic and Arctic Oceans, the figures here focus on those regions. Figures of the full global ocean climatological biases are provided in Appendix A. Observational data are the same products used in the RMSE calculation of Figure 1.

In general, the presence of the MLE parameterization improves the fidelity of the v2.1 configuration by reducing the North Atlantic Ocean surface biases present in v2 Golaz et al. (2022), as illustrated by changes to the climatological sea-surface temperature (SST; Figure 2) and sea-surface salinity (SSS; Figure 3). A reduction in the v2 sea-ice biases is also seen and is discussed in Section 3.4.

Looking in more detail at the SST field, Figure 2 shows that although the *v2.1 historical* simulation ensemble retains large-scale SST biases (Figure 2b) qualitatively similar to that of the v2 simulations (Figure 2a) – namely a meridional dipole of excess heat in the South Atlantic and a cold bias in the North Atlantic – v2.1 presents a significant SST bias reduction (blue shading in Figure 2c), focused primarily on the North Atlantic subpolar gyre, the Nordic Seas and Southern Ocean (see Appendix A, Figures A1, A2, and A3), reducing the temperature bias by 0.5-2°C. Differences for all regions, including the Southern Ocean, can be found in Appendix A.

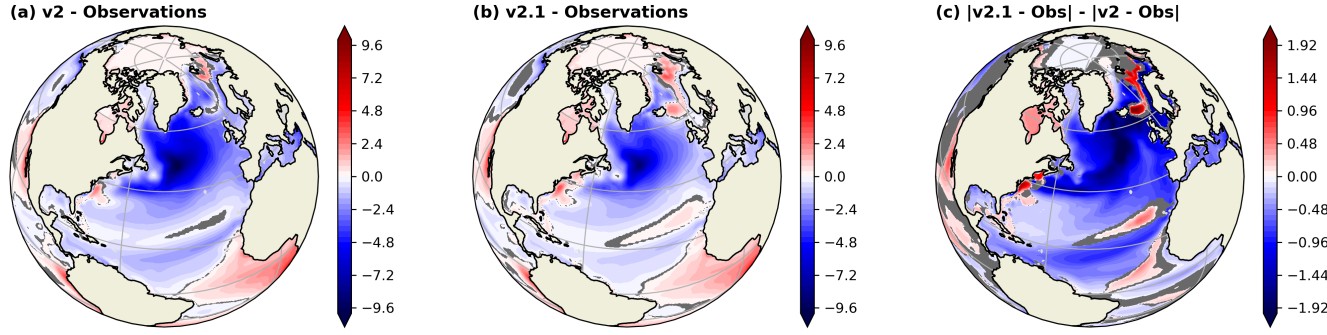

**Figure 2.** Annual climatological SST biases (°C) with respect to observations for the (a) v2 and (b) v2.1 configurations. (c) Change in SST biases between v2.1 and v2 configurations. Regions shaded in light gray denote where there is no data and regions shaded in dark gray denote where the difference is not significant (according to a one-sample T-test in (a,b) and a two-sample T-test in (c)).





Likewise, *v2.1 historical* simulation ensemble shows an improvement in the SSS bias in the North Atlantic, as shown by

the blue region in Figure 3c. The addition of the submesoscale parameterization is insufficient to fully mitigate the large scale cold, fresh bias in the North Atlantic (still visible in blue in Figures 2b and 3b), but leads to an improvement of 0.5-1 psu in the North Atlantic including both the subtropical and subpolar gyres, as well as the Nordic Seas. In the Barents, Kara, and Laptev seas, the bias reduction is even higher (1-2 psu). The v2.1 configuration and the presence of the MLE parameterization, however, do not appear to help the persistent salty SSS biases in the deep tropical North Atlantic, Beaufort and Chukchi Seas,

and Indian Ocean (see Appendix A).

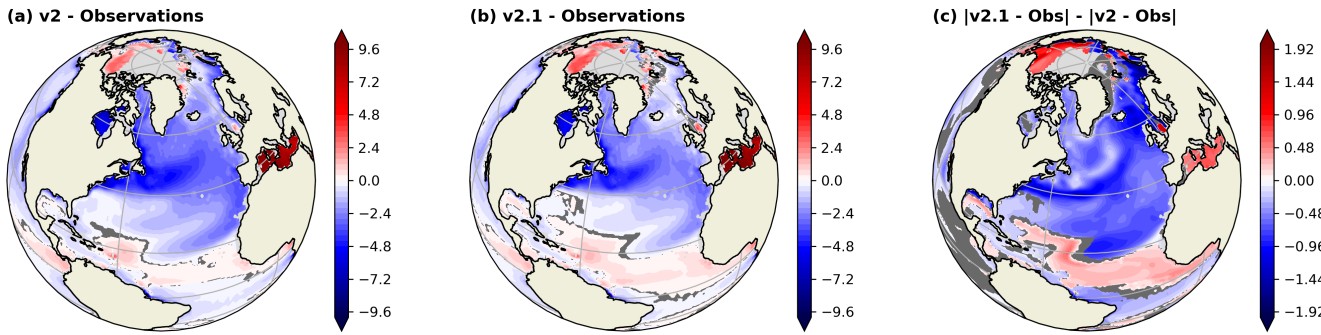

**Figure 3.** Same as Figure 2 but for SSS (psu).

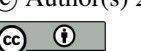

Mixed layer depths (MLD) do not show the same North Atlantic bias reduction as SST and SSS, despite having a small decrease globally (Figure 1). In particular, *v2.1 historical* simulation ensemble resulted in a slight shoaling of the mixed layer of approximately 20 meters, increasing the bias over much of the North Atlantic (Figure 4). This is most pronounced within the subpolar gyre, Nordic Seas, and eastern North Atlantic (red shading in Figure 4c). This shoaling of the MLD is one of the

primary effect of the FK11 MLE parameterization (Fox-Kemper et al., 2011)). A localized decrease in MLD does occur in the vicinity of the Gulf Stream as the v2 deep biases there are significantly reduced. This corresponds with a very small increase in the magnitude and extent of the northward limb of the western boundary current in the North Atlantic around the region of the RAPID array (Moat et al. (2019)) around 26.5 N. However, this affect is largely erased by the time the northward limb reaches the region of the west transect of the Overturning in the Subpolar North Atlantic Program array (OSNAP; (Li et al., 2021))

around the Labrador Sea, indicating no significant difference in the robustness of the subpolar gyre (not shown).

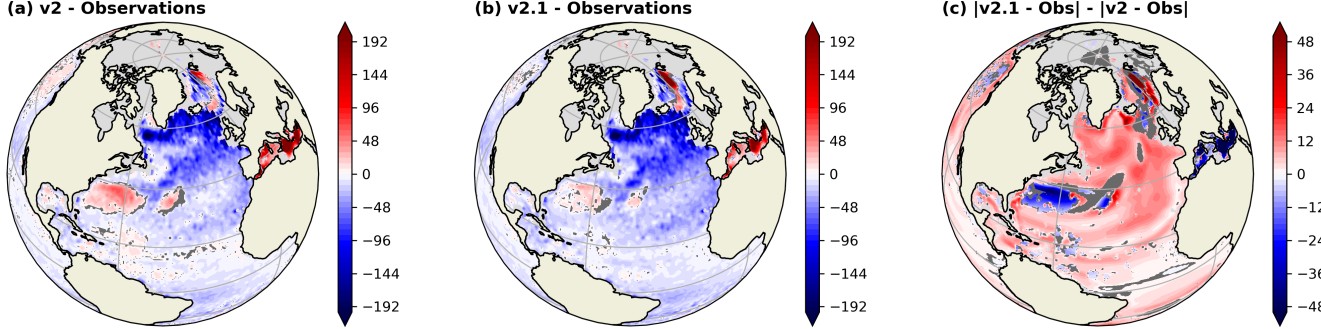

**Figure 4.** Same as Figure 2 but for MLD (m).

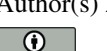



A small uptick in the strength of the AMOC is seen (Figure 5), as well as a reduction in the variability (a noted effect of the FK11 MLE parameterization (Fox-Kemper et al., 2011)). However, a larger increase in strength was expected given the reductions in the SST and SSS biases, which are likely responsible in part for reduced deep water formation in the North Atlantic that should feed into the AMOC. We explore reasons for this difference in expected versus actual outcomes in Section 4. The

bias differences in SSH or EKE are relatively small and not regionally focused, but rather somewhat uniform across the global oceans, and are therefore not shown here.

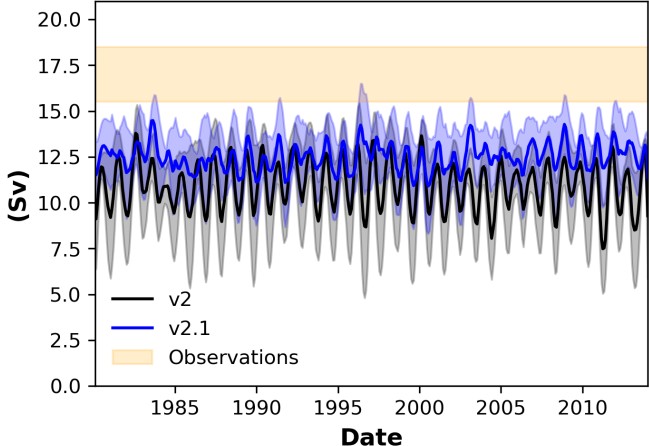

**Figure 5.** Time series of the maximum AMOC (Sv) at 26.5 N for the E3SM v2 configuration (black), the E3SM v2.1 configuration (blue), and an estimate of the observational range (orange) over the 1980-2014 period. Thick lines denote the ensemble mean, while shading illustrates one standard deviation from that mean, after a 12-month smoothing is applied.

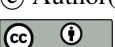


## 3.2 Atmospheric Climate

For a collective evaluation of atmospheric fields and comparison of their performance between E3SM v2 and v2.1, we applied the Program for Climate Model Diagnosis and Intercomparison (PCMDI) Metrics Package (PMP; (Lee et al., 2024)), which

is an open-source Python software package that provides quick-look objective comparisons of ESMs with one another and with observations. In Figure 6, model performances in reproducing observed global climatologies of multiple surface and upper-air fields are assessed, which include precipitation, sea level pressure, radiation at the surface and top of atmosphere, air temperature at 200 and 850 hPa, surface air temperature, surface temperature, and wind components at 200 and 850 hPa and 500 hPa geopotential height. The spatio-temporal RMSE (Gleckler et al., 2008) was calculated, which is derived by getting

a spatial RMSE of each calendar month of climatological annual cycle and then averaging it across months. Although the differences are small, Figure 6 indicates most fields in v2.1 have a reduced bias compared to v2. The improvement in surface temperature ("ts") and near-surface air temperature ("tas") is noticeable, which may be dominated by the improvement of the sea-surface temperature field as shown in Figure 2.

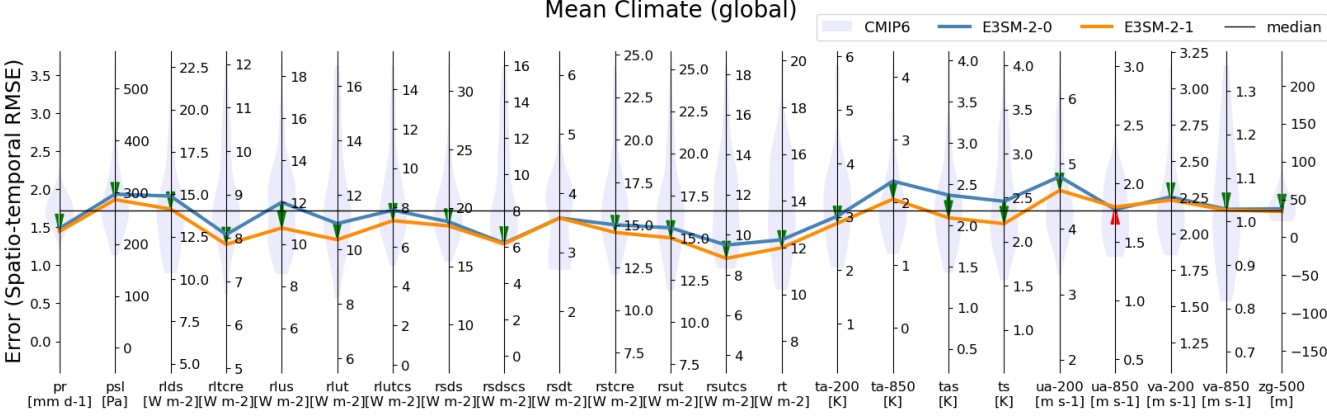

**Figure 6.** The PMP Parallel Coordinate Plot (Lee et al., 2024) for global mean climate evaluation, showing the spatio-temporal RMSE (Gleckler et al., 2008). Each vertical axis represents a different variable. Full names for model variables on the abscissa and their reference datasets can be found in Table 1 of Lee et al. (2024). RMSEs are constructed using the five *v2/v2.1 historical* ensemble member over a 1981-2005 time period. Improvement (degradation) in E3SM v2.1 compared to E3SM v2 is highlighted as a downward green (upward red) arrow between lines. The midpoint of each vertical axis is shifted to represent the median result from the CMIP6 multi-model ensemble (horizontal black line), with the axis range stretched to the minimum and maximum from the median CMIP6 for visual consistency. The inter-model distributions of CMIP6 model performance are shown as shaded violin plots along each vertical axis. First historical ensemble member of each model is used for the assessment.

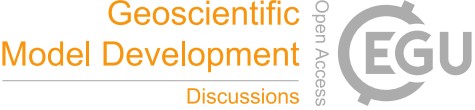

### 3.3 Extratropical Modes of Variability

To examine the influence of the model update to the interannual climate variability modes, we applied the PMP's metrics for extratropical modes of variability (ETMoVs) for E3SMv2.1 and v2. Specifically, we have evaluated five atmospheric-based ETMoVs including the Northern Annular Mode (NAM), North Atlantic Oscillation (NAO), Pacific North America pattern (PNA), North Pacific Oscillation (NPO), and Southern Annular Mode (SAM), as well as two based on sea-surface temperature (SST): the Pacific Decadal Oscillation (PDO), and North Pacific Gyre Oscillation (NPGO). The atmospheric ETMoVs were

evaluated for four seasons, while monthly time series were directly used for the SST-based modes. The Common Basis Function (CBF) approach is used to ensure a fair comparison of ETMoVs as simulated by climate models (Lee et al., 2019, 2021, 2024). The metric we selected for this study is the spatio-temporal RMSE obtained from the comparison of the model's CBF pattern to the observed empirical orthogonal functions (EOF) pattern, which enables the inter-model comparison of bias magnitude that is from both the pattern and amplitude (Lee et al., 2019). To gauge the influence of internal variability in the evaluation process,

we use the five *v2/v2.1 historical* ensemble members. Metrics were calculated for each ensemble member of the model, then averaged across the five realizations. In Figure 7, there are 14 metrics that show improvement including NAM and NAO and 8 cases of degradation. The ETMoV performance has not substantially changed in the context of the inter-model spread of CMIP6 models (shown as purple violin plot in the background). In summary, the large-scale extratropical modes of variability are not substantially different between E3SM v2 from v2.1.

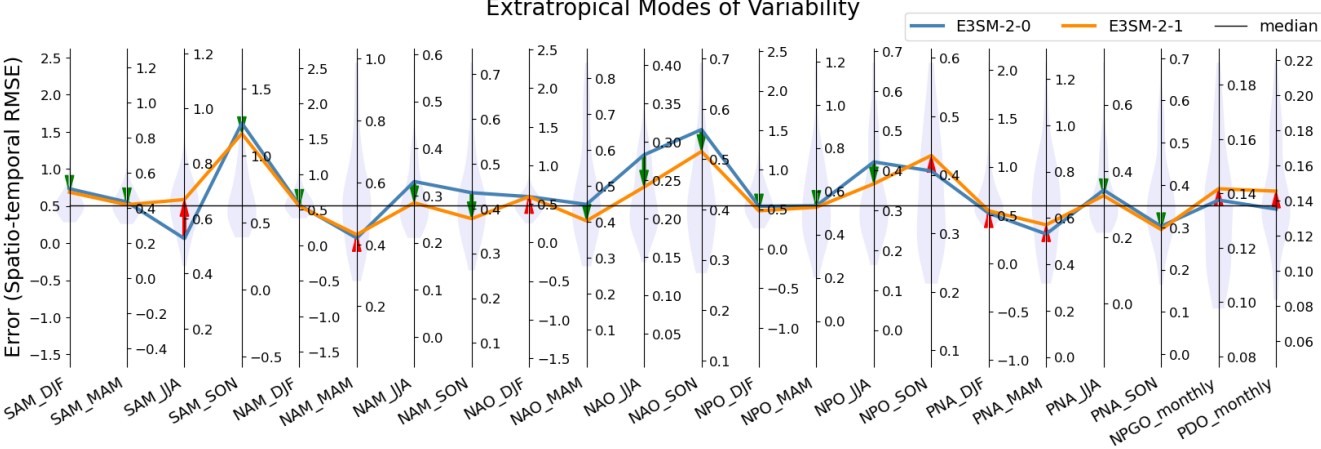

**Figure 7.** The PMP Parallel Coordinate Plot (Lee et al., 2024) for extratropical modes of variability evaluation, showing the spatio-temporal RMSE (Lee et al., 2019). Each vertical axis represents a different mode and season. Analysis is constructed using the five *v2/v2.1 histori-cal* ensemble members over a 1900-2005 time period. Improvement (degradation) in E3SMv2.1 compared to E3SMv2 is highlighted as a downward green (upward red) arrow between lines. Similar to Figures 6, the middle of each vertical axis is set to the median of the group of benchmarking models (i.e., CMIP6), with the axis range stretched to maximum distance to either minimum or maximum from the median for visual consistency. The inter-model distributions of model performance are shown as shaded violin plots along each vertical axis.





## 3.4 Sea-Ice Climate


Similar to Section 3.1, in Figures 8 and 9 we plot climatological means of sea ice properties constructed using the five member *v2/v2.1 historical* ensemble means over a 1980-2014 period. Observational data are derived using measurements from multiple sensors across many satellite platforms detailed in Comiso (2017). Figure 8 shows an improvement in v2.1 Arctic sea-ice concentration, particularly around the Nordic Seas, Greenland, and the Labrador Sea (blue shading in Figure 8c). Figure 9

shows a mix of improvement in the Indian Ocean and degradation in the Weddell, Amundsen, and Ross Seas for v2.1 Antarctic sea-ice concentration.

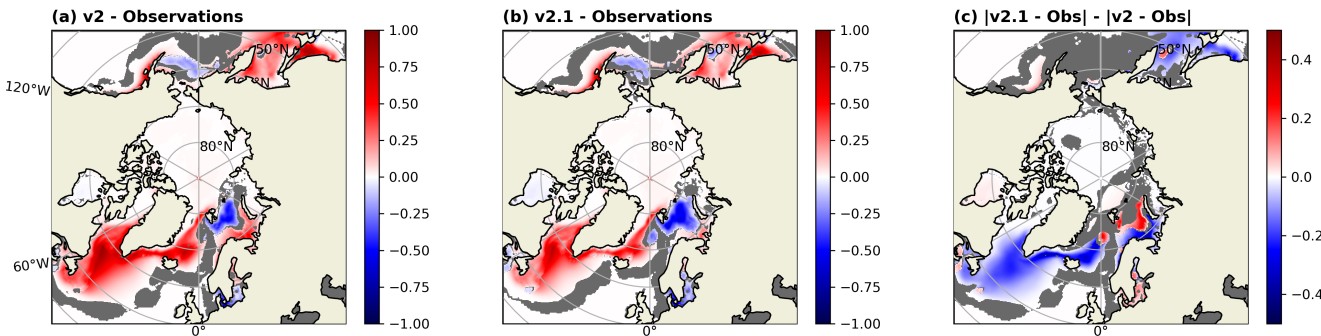

**Figure 8.** Annual climatological sea-ice concentration (fraction) biases with respect to observations in the Arctic for the (a) v2 and (b) v2.1 configurations. (c) Change in sea-ice concentration biases between v2.1 and v2 configurations. Regions shaded in light gray denote where there is no data and regions shaded in dark gray denote where the difference is not significant (according to a one-sample T-test in (a,b) and a two-sample T-test in (c)).

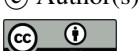



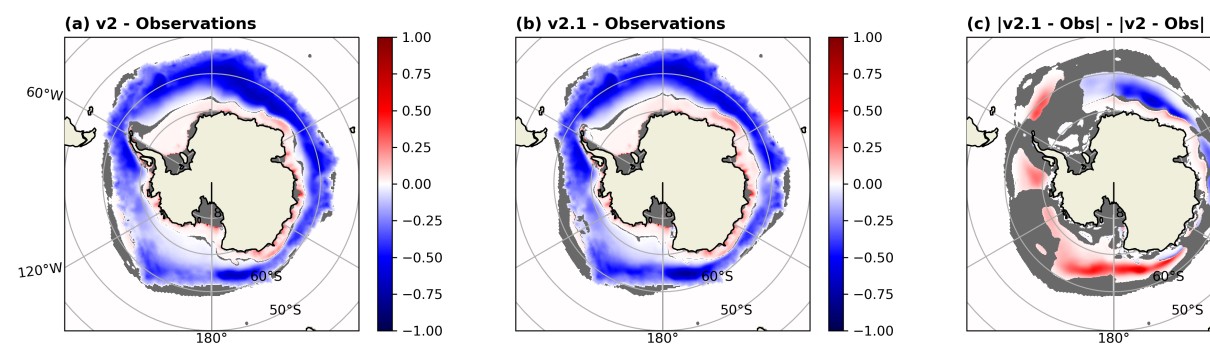

**Figure 9.** Same as Figure 8 but for the Antarctic.





We evaluated the time mean and climatological annual cycle of sea-ice area in both the Arctic and Antarctic. In the calculation of the metric, we defined sea-ice area following Ivanova et al. (2016) as the area of grid cells with greater than 15% sea ice coverage (i.e. 15% ice concentration) multiplied by their fraction of coverage and summed across grid cells within each region.

For each of the Arctic and Antarctic regions, we partitioned the geographical region into three domains following Ivanova et al. (2016). The Arctic region is separated into the North Pacific, Central Arctic, and North Atlantic (shown in Figure A5 in Appendix A), and the Antarctic region is separated to the South Atlantic, Indian Ocean, and South Pacific domains (shown in Figure A6 in Appendix A). The model output is evaluated against the EUMETSAT OSI-SAF satellite based product (Ocean and Facility, 2022). A date range of 1988-2014 was chosen for the maximal overlap between the *v2/v2.1 historical* simulations

and periods without missing data in the OSI-SAF product. The sea-ice area metrics that were proposed by Ivanova et al. (2016) and recently implemented in the PMP (Lee et al., 2024) are used for the analysis. We derived a simulated annual cycle and annual mean of the sea-ice area in each region for the time period of 1988-2014, to compare with observations (Figure 10). The evaluation metrics are defined as mean square errors in the the climatological annual cycle after removing the annual mean, and annual mean of the sea ice area (Figure 11). The overestimation of sea-ice area over the North Atlantic, North Pacific (mostly

in December to May, Figure 10) and the Central Arctic (in August to September, Figure 10), and the underestimation over the Indian Ocean (in July to October, Figure 10) sub-regions in v2 are noticeably alleviated in v2.1, while the changes in other regions do not appear to be substantially different.





**Figure 10.** Climatological annual cycle (1988-2014) of total sea-ice area in the entire Arctic and its sub-regions (left) and the entire Antarctic and its sub-regions (right), obtained from the E3SMv2 (blue) and v2.1 (orange) simulations, and the reference dataset EUMETSAT OSI-SAF (black line). Solid lines indicate the average of the five ensemble members while the shaded color indicates inter-ensemble spread. Dashed horizontal lines indicate the respective annual mean of the sea-ice area. In Antarctic panel, the blue dashed line is hidden beneath the orange dashed line.





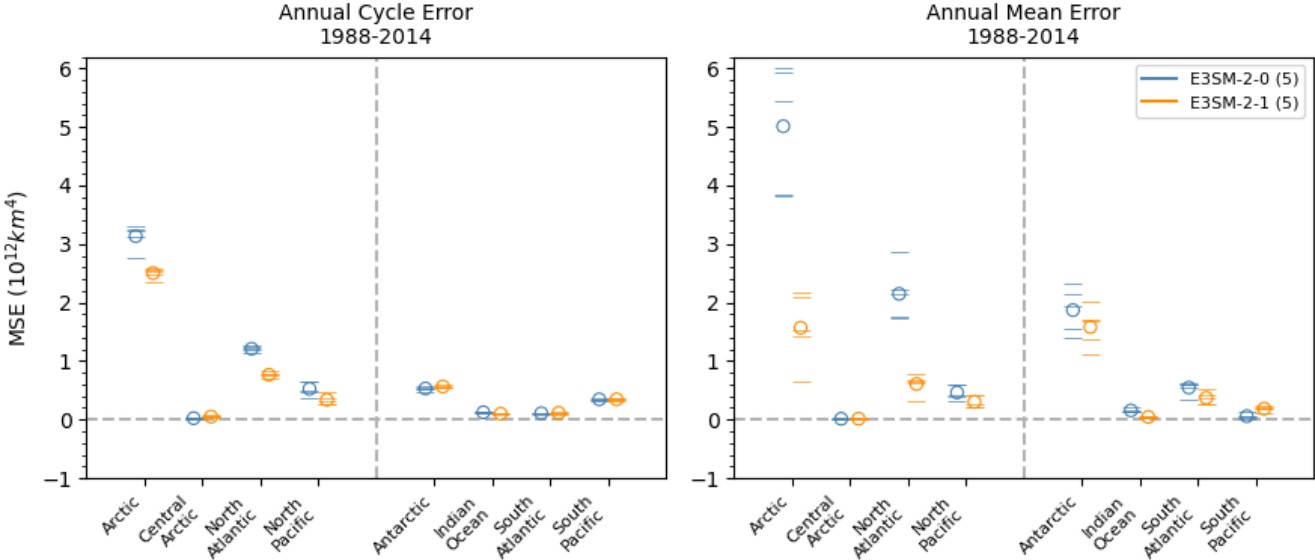

**Figure 11.** Mean square error (MSE) of the total sea-ice area annual cycle (annual mean removed; left) and annual mean (i.e., time-mean bias squared; right) of each Arctic and Antarctic sub-region. Horizontal line markers indicate the metric obtained from individual ensemble members and open circle markers indicate the multi-realization averages of the five ensemble members for each sub-region. Lower values are better. The gray vertical dashed line separates sub-regions of Arctic (left) and Antarctic (right) in each panel. The diagnosed annual means and cycles for all regions, simulations, and the reference dataset can be found in Figure 10.





### 3.5 Climate Sensitivity and Transient Climate Response

We also performed two idealized $CO_2$ simulations (*v2/v2.1 1pctCO2* and *v2/v2.1 abrupt-4xCO2*) to estimate the model sensi-
tivity to $CO_2$-forcing at different time scales. The equilibrium climate sensitivity (ECS) is defined as the equilibrium surface
temperature change resulting from a doubling in $CO_2$ concentrations. ECS is typically approximated by linear regression of
top-of-atmosphere radiation vs. surface temperature in a 150 yr "abrupt-4xCO2" simulation (Gregory et al., 2004), often re-
ferred to as "effective climate sensitivity". Response on shorter time scales is measured by the transient climate response (TCR)
defined as the change in surface temperature averaged for a 20 year period around the time of $CO_2$ doubling from a "1pctCO2"
simulation. TCR depends on both climate sensitivity and ocean heat uptake rate.

ECS is nearly unchanged in E3SMv2.1 at 3.92 K compared to 4.0 K in E3SMv2. TCR is slightly smaller at 2.20 K compared
to 2.41 K. Both values are substantially smaller than E3SMv1 which suffered from a high sensitivity (ECS of 5.30 K and TCR
of 2.93 K; (Golaz et al., 2019)). Meehl et al. (2020) evaluated ECS and TCR for 37 CMIP6 models. They found that the
multimodel mean ECS was $3.7 \pm 1.1$ K (standard deviation). The multimodel mean TCR was $2.0 \pm 0.4$ K. E3SMv2.1 is well
within one standard deviation of multimodel mean for both ECS and TCR, but higher than the mean.

In the next section, we delve into the potential mechanisms for the changes in biases present in the North Atlantic Ocean, as
well as potential links between those biases and AMOC.

### 4 Relationship between AMOC and North Atlantic state

In order to shed light on mechanisms creating these changes, in particular the relationship between AMOC and the North At-
lantic biases, we examined several variables in detail. This was done in an effort to relate deep water formation and overturning
to surface biases. To do this, we primarily investigated tracer transports through the North Atlantic via a 33-year *piControl*
extension run for both the v2 and v2.1 configurations (*v2/v2.1 piControl Ext*), with three additional passive tracers. At year
501 of both the v2 and v2.1 *piControl* simulations, we added three passive tracers and ran the simulations out to year 533.
The injection of one tracer was proportional to the sea-ice fresh water flux into/out of the ocean, the second was set to one
in the first grid layer globally at every time step, and the last was set to one in the first grid layer at every time step, but only
within a North Atlantic latitude and longitudinal band of $50° - 80°$N and $60°$W$-10°$E, respectively. Conclusions from each
of these tracers were roughly the same, so for clarity we only show and discuss the third tracer. For consistency between plots
and analysis in this section, all figures and analysis utilize these *v2/v2.1 piControl Ext* runs with passive tracers, instead of the
ensemble *v2/v2.1 historical* runs utilized in the previous sections.

Figure 12 shows the annual climatological (a-b) MLD and (d-e) sea-ice concentration in the Nordic seas for the (a,d) v2 and
(b,e) v2.1 configurations, and changes in (c) MLD and (f) sea-ice concentration between the two configurations. The MLD
and sea-ice concentration plots in Fig. 12 here differ from those in Figures 4 and 8, in that Figures 4 and 8 show (a,b) biases
with respect to observations and (c) the change in biases with respect to observations, while Fig. 12 shows just (a,b,d,e) total
MLD and sea-ice concentration and (c,f) the differences in total MLD and sea-ice concentration between the v2 and v2.1





configurations. Thus, pattern and maximum differences in Figs. 4(c), 8(c), and Fig. 12(c,f) are not expected to be in exact agreement with each other.

While v2.1 showed slightly shallower MLDs than v2, both globally and in the subpolar gyre (outlined in the previous section), different patterns emerge in the Nordic Seas (Fig. 12 a-c). In particular, we see a substantial deepening of the MLD by several hundred meters south of Svalbard and the Fram Strait in v2.1 compared with v2. A similar global shoaling and

localized Nordic Seas deepening of the mixed layer was seen in both the Nucleus for European Modelling of the Ocean (NEMO) model (Calvert et al., 2020) and several coupled Earth system models in Fox-Kemper et al. (2011). Due to the emergence of this behavior only in fully coupled models and not in ocean-only models driven by normal year forcing, Fox-Kemper et al. (2011) attributed this response to air-sea and ice-sea feedback triggered by the MLE parameterization. In E3SM, while northern hemisphere sea ice concentration is still too large in both model configurations, it was less so in v2.1 (Fig. 8

and 12 d-e), resulting in a climatologically more open ocean in the Nordic Seas. Along with increased vertical mixing and surface fluxes within that same region (not shown), we speculate that similar air-sea and ice-sea feedbacks triggered by the MLE parameterization are leading to the greater Nordic Seas MLDs in v2.1.

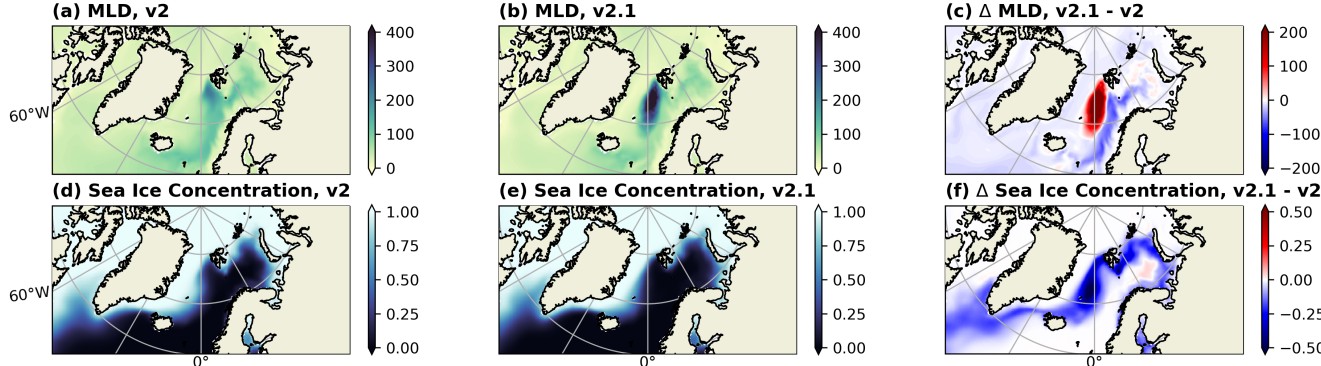

**Figure 12.** Annual climatological (a-b) MLD (m) and (d-e) sea-ice concentration (fraction) in the Nordic Seas for the (a,d) v2 and (b,e) v2.1 configurations. Change in (c) MLD and (f) sea-ice concentration between the v2.1 and v2 configurations.



To further understand the effects of these feedbacks and deeper MLDs, we examined several passive tracers. Figure 13 shows the North Atlantic passive tracer (a-d) concentration for the v2.1 configuration at various depths and (e-h) the change

in concentration between the v2.1 and v2 configurations at those same depths. Most notably, panels (d) and (h) show tracer concentrations at depths greater than 1000m within the Nordic Seas and along the edge of the Nansen Basin are much higher in the v2.1 configuration than the v2 configuration. Combined with enhanced MLDs and mixing and reduced sea-ice coverage in Nordic Seas, the v2.1 configuration appears to form significantly more deep, dense water in this region that is then exported into the Arctic. By forming more deep water here that is exported to depth by vertical mixing, the cold, fresh biases present at

the surface in v2 are reduced in v2.1.

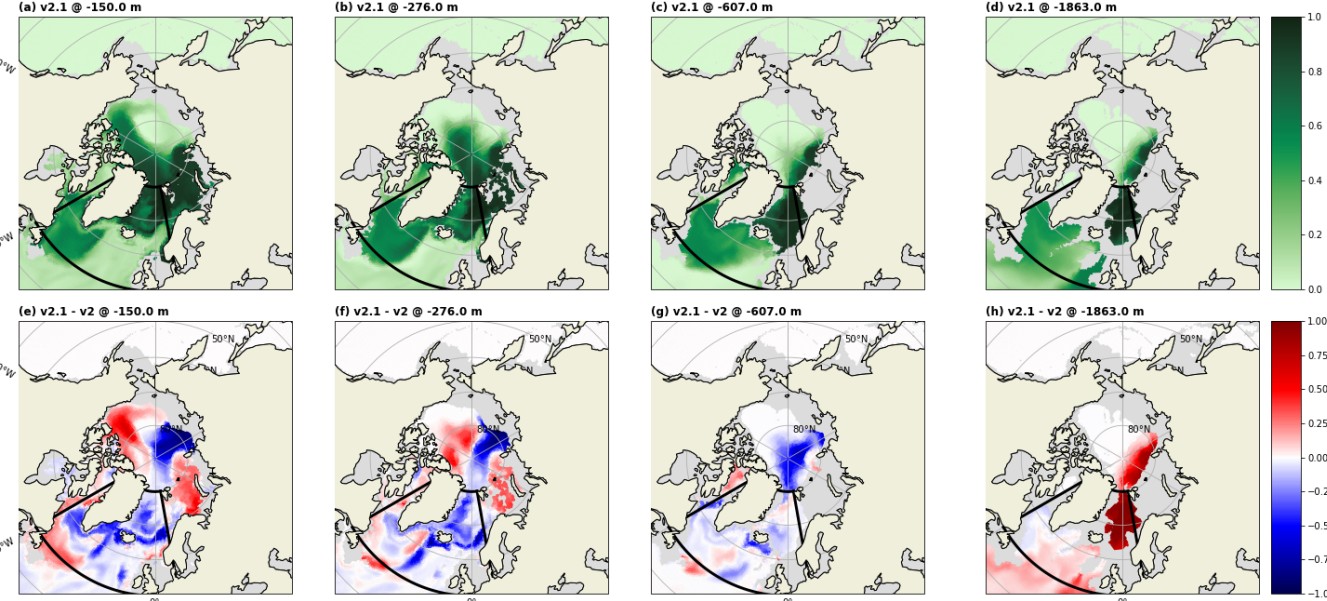

**Figure 13.** (a-d) Tracer concentration in the v2.1 configuration and (e-h) the change in concentration between the v2.1 and v2 configurations 33 years after tracer initiation at depths of (a,e) -150m, (b,f) -276m, (c,g) -607m, and (d,h) -1863m. Thick black boxes indicate the extent of the tracer surface forcing and light gray shading denotes bottom topography at that model depth.





However, the vast majority of this extra deep water is exported to the Arctic via North Atlantic Current flow through the Fram Strait (∼80% of the released tracer is north of 75°N at the end of the 33 years in both v2 and v2.1). Very little ends up below 1000m in the Atlantic, resulting in little change in the overturning circulation and providing a potential explanation for the lack of notable increase in the AMOC strength from v2 to v2.1. Figures 14 and 15 show changes in the winter and summer

temperature and salinity along transects within the Labrador and Nordic Seas between the v2 and v2.1 configurations. Within the Labrador Sea, where deep water is typically formed to feed the AMOC, we see very little change in the density/stratification structure between v2 and v2.1. A strong stratification buffer remains near the surface, preventing export of the cold, fresh water that remains there. Conversely, within the Nordic Seas, we see most of the interior stratification (below 1000m) has been eroded away by the full-depth mixing associated with the deeper MLDs there. This full-depth mixing is present for most of the 500

year *v2.1 piControl* simulation and appears to have developed very early on in both the Greenland and Norwegian Seas (see Figure A4 in Appendix A). This lack of interior stratification and full-depth mixing allows for a greater connection of the surface to the deep, leading to greater export.

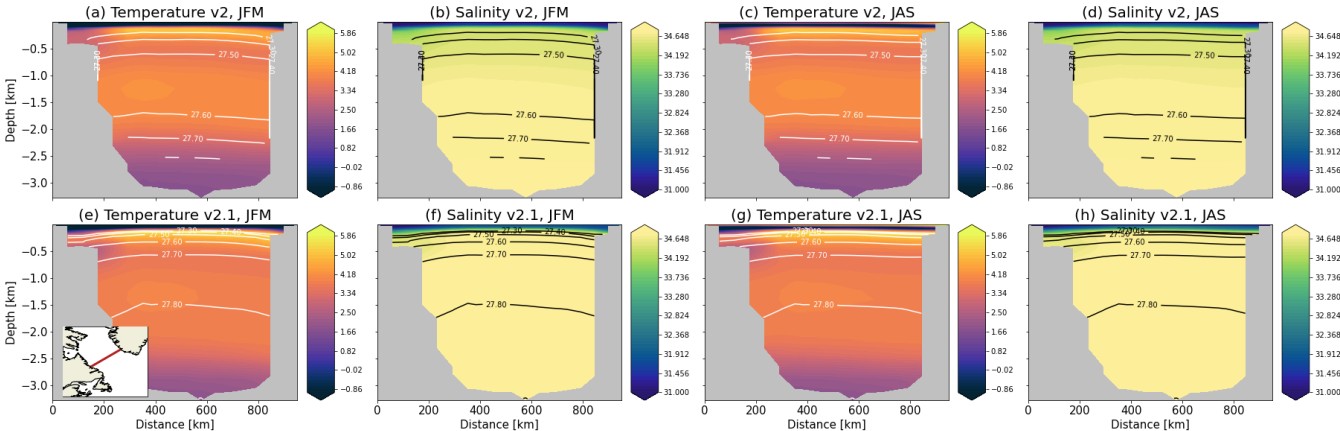

**Figure 14.** Annual climatological (a,c,e,g) temperature and (b,d,f,h) salinity in the (a,b,e,f) winter [JFM] and (c,d,g,h) summer [JAS] for the v2 (top row) and v2.1 (bottom row) configurations across a Labrador Sea transect (transect location shown in panel (e) inset). Black contour lines show select values of constant potential density.



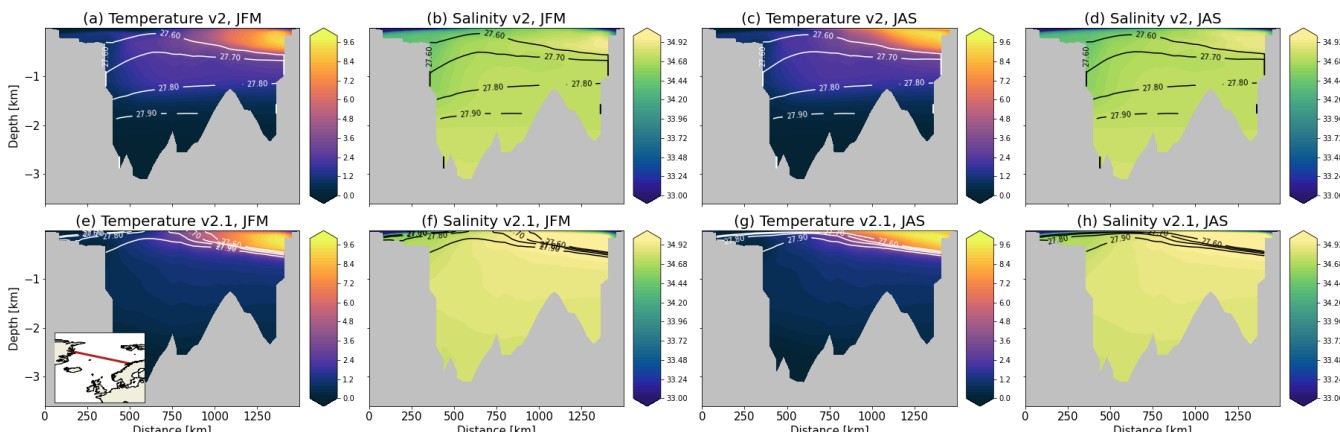

**Figure 15.** Same as Figure 14 but for a Nordic Seas transect.





Figure 16 further examines the formation of this deep water, by showing the surface water mass transformation in the North Atlantic basins for v2 (solid line) and v2.1 (dashed) over the *v2/v2.1 piControl Ext* runs. The diagnosed surface water mass
transformation highlights a clear difference with the added MLE parameterization. In most regions of the North Atlantic and Nordic Seas, introducing the MLE parameterization in v2.1 leads to the formation of more dense waters, either from an increase in the density of the waters formed (a shift to the right in Fig. 16) and/or from an increase in the dense water formation rate (visible as an increase in the peak value of the transformation rate). Specifically, in the v2.1 configuration, the North Atlantic Ocean shows a transformation rate peak (17.8 Sv) similar in magnitude to that of v2 (18 Sv), but shifted to higher densities:
the range of dense water formed (visible as a negative slope in transformation) is 26 - 27.2 kg m$^{-3}$ instead of the lighter 25.6 - 27 kg m$^{-3}$ range formed in the v2 configuration. Similarly, the dense water formation in the Norwegian Sea is shifted from 27.1 - 27.6 kg m$^{-3}$ to 27.4 - 27.9 kg m$^{-3}$ with the addition of the MLE parameterization (with its magnitude remaining 3.9 Sv across both configurations). In other regions such as the Iceland and Greenland Seas, both the peak transformation rate and the density range have increased in v2.1. In the Iceland basin, the surface fluxes now form 2.3 Sv of dense water (27.1 - 27.7
kg m$^{-3}$) instead of the weaker 1.5 Sv of slightly lighter waters (26.9 - 27.5 kg m$^{-3}$) in v2. In the Greenland Sea, they form 1.3 Sv of dense water (27.8 - 28 kg m$^{-3}$) compared to the 0.7 Sv of lighter waters (27.6 - 27.8 kg m$^{-3}$) formed in v2 – a near doubling of dense water formation in that region.

In the Labrador and Irminger Seas, the transformation rates and changes to transformation are small compared to the regions mentioned above. It is important to note that these last two regions are considered important sites of convection and dense water
mass formation in observations: convection in the Labrador Sea leads to important subpolar mode water formation, while deep water formation in the Irminger Sea is considered a major contributor to Atlantic Deep Water, AMOC (Petit et al., 2020), and possibly Labrador Sea Water (Pickart et al., 2003). Representing dense water formation in climate models in the correct locations and appropriate magnitudes is an ongoing challenge, given that small-scale physical processes that contribute to it (e.g. convection, eddy mixing) are parameterized in models.



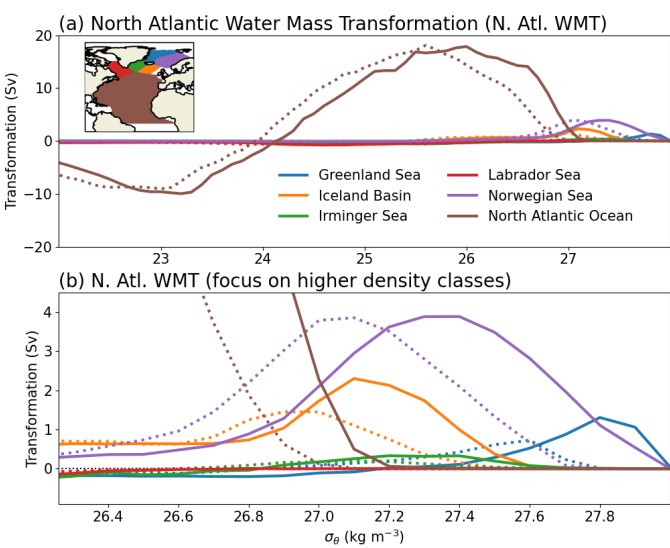

**Figure 16.** Surface water mass transformation in the North Atlantic basins in the v2.1 (solid) and v2 (dotted lines) configurations over the 33-year extension (years 501-533). Panel (a) shows the full analyzed density range, while panel (b) zooms in on the higher density classes relevant to the specific basins and seas examined in this paper. Scope of each region is shown in the map inset in panel (a).





## 5 Conclusions

Overall, the v2.1 configuration resulted in some improvements to North Atlantic ocean biases, particularly SST and SSS (Figures 2 and 3), which resulted in improved sea-ice concentration in the North Atlantic (Figure 8). Changes to MLD biases were as expected, mirroring FK11 with a general shoaling of the mixed layer (Fox-Kemper et al., 2011) visible in many regions (Figure 4), though coupled feedbacks lead to regional deepening of the MLD that reduced the mean MLD bias globally (Figure 1). Additionally, though MLD is underestimated in comparison to observations in most regions, it does not appear to lead to any widespread degradation in the climate.

AMOC magnitude increased slightly with the addition of the MLE parameterization (Figure 5). This may be expected, since the direct effect of the FK11 parameterization on AMOC appears to be model-dependent and tends to either stabilize or minimally affect AMOC (Fox-Kemper et al., 2011). This effect holds for the v2.1 ensemble of simulations, which showed less variability in AMOC overall. Differences in AMOC may be due at least in part to a relocation of the site of deep convection from the Labrador and Irminger Seas and North Atlantic to the Nordic Seas (Fox-Kemper et al., 2011). This hypothesis aligns with changes we saw to our modeled MLDs in v2.1 when compared with v2.

There appear to be differences in MLD in the Nordic Seas that are separate from the overall MLD shoaling created in v2.1, indicating potential changes in deep water formation there (Figure 12). Similar to Fox-Kemper et al. (2011), we also saw deep convection occurring in the Nordic Seas rather than in the Labrador and Irminger Seas. Our water mass transformation analysis shows changes in deep water formation, particularly in the North Atlantic and Norwegian Sea, with more dense water being formed in v2.1 than in v2 (Figure 16). While it's not clear why this is the case, it does explain why our modeled AMOC improves slightly while not drastically altering the North Atlantic Subpolar Gyre. If increased deep water formation was occurring in the Labrador and Irminger Seas, we would expect to see a great decrease in the SSS, SST, and sea-ice biases there. The passive tracer transport analysis in Section 4 for v2.1 indicates that the increased deep convection may not translate to a better AMOC because the dense water formed is transported northward into the Arctic, rather than towards the south. Comparison of tracer advection from v2 to v2.1 shows deep northward convection from the Barents Sea into the Arctic in v2.1; this does not occur in v2. Future work will attempt to elucidate mechanisms behind this shift in deep water formation.

Our results also indicate that the climatological characteristics of many surface and atmospheric fields in E3SMv2.1 have improved compared to v2 (Figure 6). We have also examined the model performance of seven inter-annual extratropical modes of variability, including the atmospheric-based NAM, NAO, PNA, NPO, and SAM, as well as two modes based on sea-surface temperature (SST): the PDO, and NPGO. Our results suggest noticeable improvement in the NAO and NAM, however, for most modes and seasons we find the large-scale extratropical modes of variability in E3SMv2.1 are not significantly different from v2 (Figure 7). Lastly, our analysis indicates that the TCR and ECS are essentially unchanged in E3SMv2.1 compared to v2.





*Code and data availability.* The E3SM code is available at https://github.com/E3SM-Project/E3SM, and the model versions used for the simulations presented here are E3SM v2.1 and v2. A full list of all code changes made from v2 to v2.1 can be found here: https://github.com/E3SM-Project/E3SM/pulls?q=is%3Apr+is%3Aclosed+merged%3A2021-09-29..2023-01-11+base%3Amaster+label%3Anon-BFB+. Information about running the model is available at https://e3sm.org/model/running-e3sm. The simulation data used for this paper is pub-
310   lished to CMIP6 through the Earth System Grid Federation (ESGF). Data is available from https://aims2.llnl.gov/search/cmip6/ under source_id=E3SM-2-1. Preliminary analysis on the ocean component MPAS-Ocean was performed using MPAS-Analysis, available at https://github.com/MPAS-Dev/MPAS-Analysis (doi 10.5281/zenodo.4407459).

## Appendix A: Supplementary Figures

**(a) v2 - Observations**

**(b) v2.1 - Observations**

**(c) |v2.1 - Obs| - |v2 - Obs|**

**Figure A1.** Global annual climatological SST biases (°C) with respect to observations for the (a) v2 and (b) v2.1 configurations. (c) Change in SST biases between v2.1 and v2 configurations. Regions shaded in light gray denote where there is no data and regions shaded in dark gray denote where the difference is not significant (according to a one-sample T-test in (a),(b) and a two-sample T-test in (c)).





**Figure A2.** Same as Figure A1 but for SSS (psu).



**Figure A3.** Same as Figure A1 but for MLD (m).



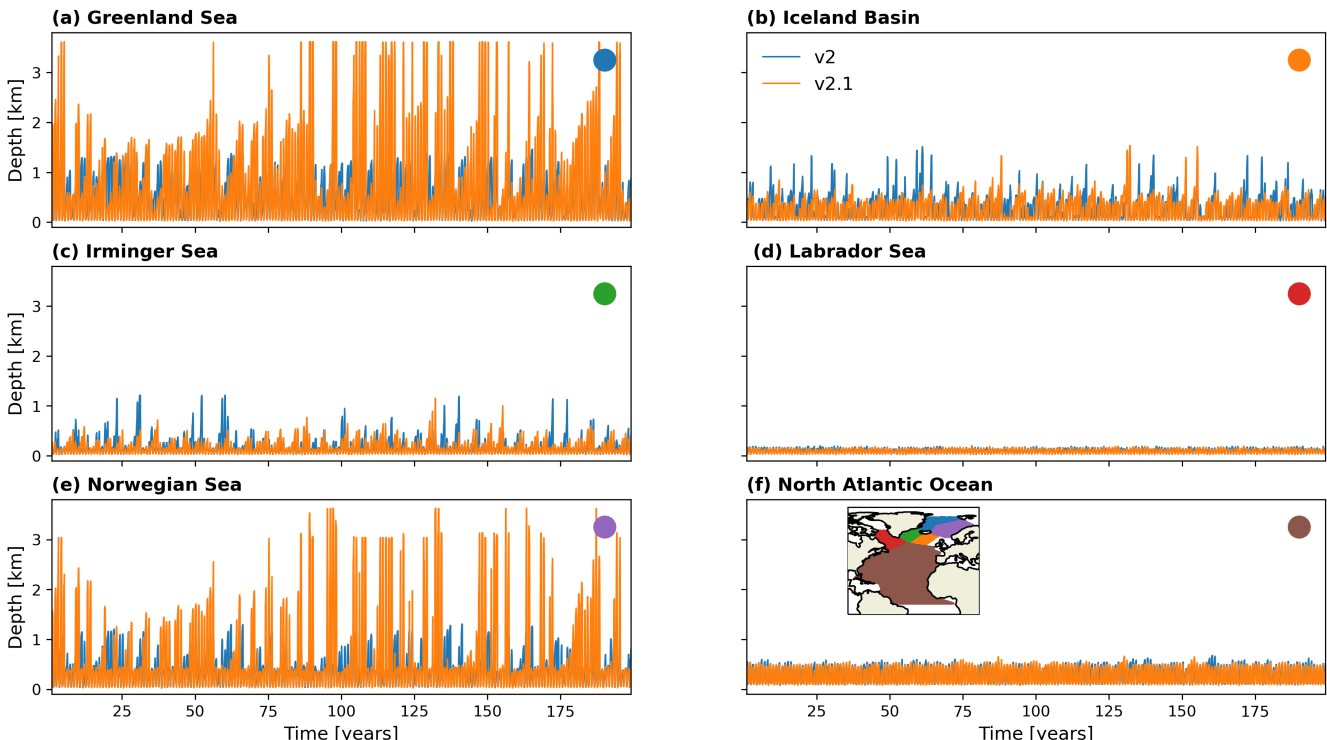

**Figure A4.** Maximum mixed layer depth (m) for the first 200 years of the *v2 piControl* (blue lines) and *v2.1 piControl* (orange lines) in the North Atlantic basins: the (a) Greenland Sea, (b) Iceland Basin, (c) Irminger Sea, (d) Labrador Sea, (e) Norwegian Sea, and (f) North Atlantic Ocean. Scope of each region is shown in the map inset in panel (f) and correspond to the same regions used in Figure 16.





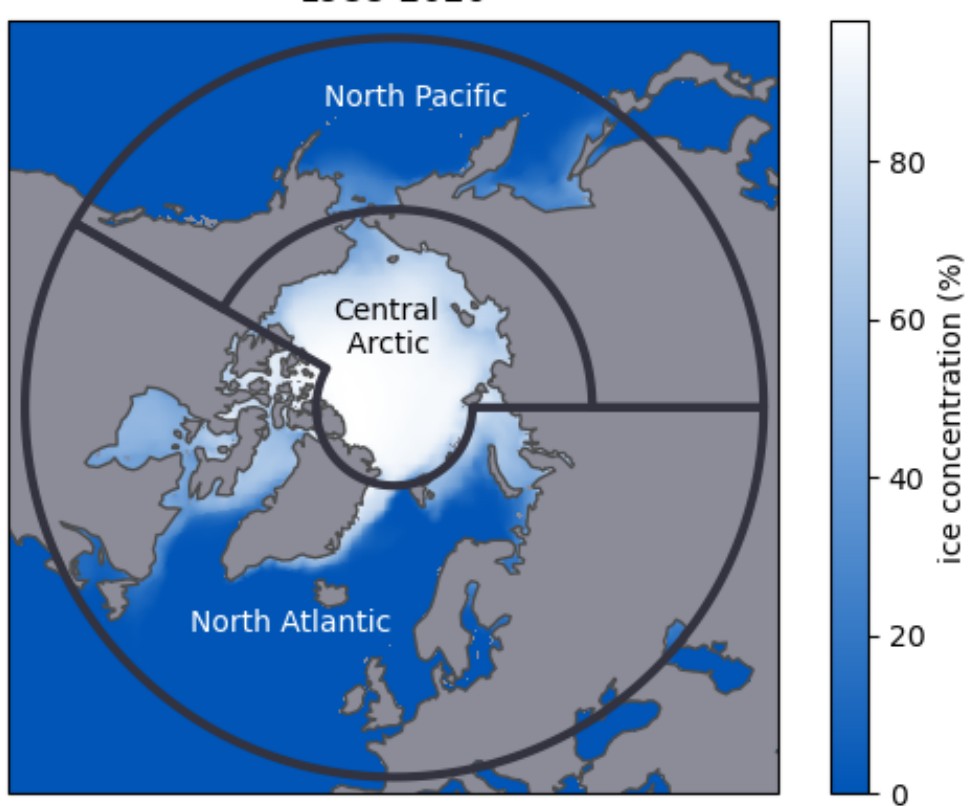

**Figure A5.** Partitioned Arctic geographical regions for the sea-ice area analysis in Section 3.4 Figures 10 and 11 (Ivanova et al., 2016).

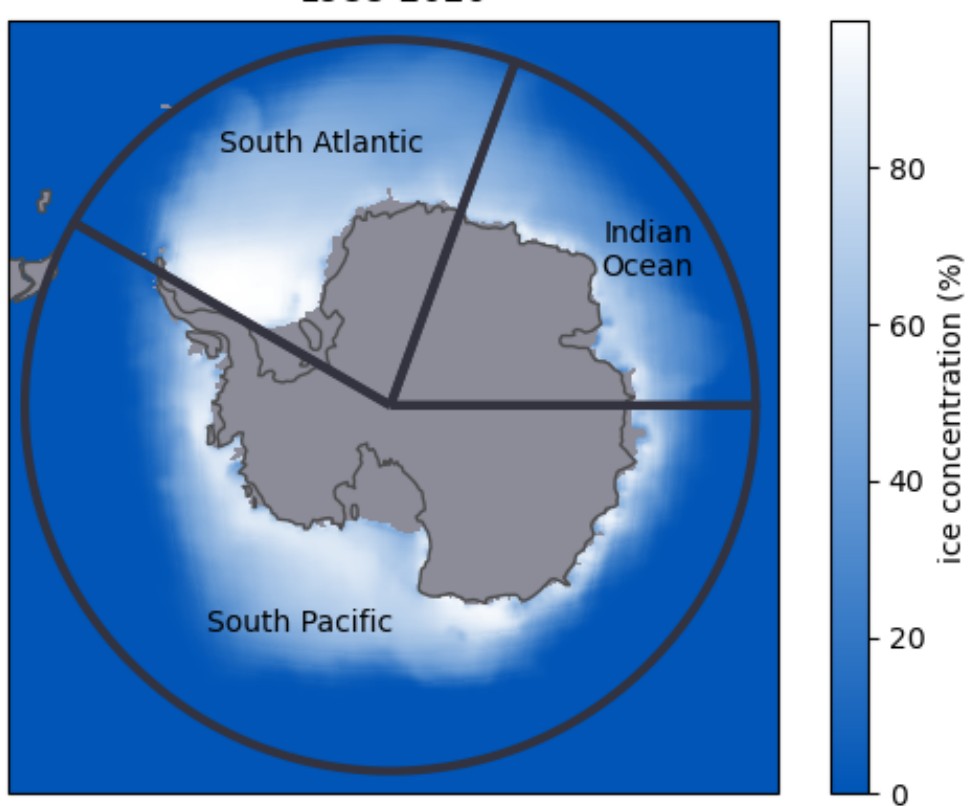

**Figure A6.** Same as Figure A5 but for the Antarctic.





## Appendix B: Code Changes from v2 to v2.1

### B1 Atmosphere

The atmosphere component of E3SM remains the E3SM Atmosphere Model (EAM). While there were no major changes in the default configuration of EAM from version 2, several new features are included in the base code including: a semi-Lagrangian tracer transport for theta-l dycore; a new algorithm for finding the tropopause; and new regionally refined mesh configurations.

### B2 Ocean

The ocean component of E3SM remains MPAS-Oean. Beyond the FK11 MLE parameterization discussed in detail in the main part of this manuscript, a correction for barotropic thickness consistency that reduces divergence noise and the addition of an ocean carbon conservation analysis member were made.

### B3 Sea Ice

The sea ice component of E3SM remains MPAS-Seaice. From v2 to v2.1 a correction to how shortwave parameters are interpolated in the SNICAR-AD 5-band radiation scheme, the addition of a sea ice carbon conservation analysis member, updates to the default sea ice biogeochemistry namelist parameters to be consistent with v2 improvements to nitrogen cycling, and a correction in the ice-ocean dissolved organic nitrogen coupling were made.

### B4 Land

The land component of E3SM remains the E3SM Land Model (ELM). While there were no major changes in the default configuration of ELM, several optional features have been added including; the implementation of topography-based subgrid structure and accompanying parameterizations and atmopsheric forcing downscaling; a new plant hydrolics scheme; tw0-way land-river hydrological coupling through the infiltration of floodplain water; and implementation of perennial crops; updates to the SNICAR-AD snow radiative transfer model; and implementaiton of soil erosion and sediment yeild in ELM-Erosion.

### B5 River

The river component of E3SM remains the Model for Scale Adaptive River Transport (MOSART). While there were no major changes in the default configuration of MOSART, an optional feature of two-way river-ocean hydrological coupling between MOSART and MPAS-Ocean is included.

### B6 Coupled System

The coupler in E3SM remains cpl7 Craig et al. (2012). Small bug fixes in the land-atmosphere fluxes and in the zenith angle calculation were made, along with the option to calculate carbon budgets when heat and water budgets are active.



*Author contributions.* KS, AB, LC, LVR, CB, MP, KB, OG, JW, MM, WL, and AH contributed to the improvements of AMOC and bias representation in E3SM. PG, JL, AO, and CZ are model evaluation contributors to the PCMDI Metrics Package, and PG, JL, and AO provided the atmosphere and part of the sea-ice analysis and text. JCG and LVR ran the v2.1 simulation campaign. CB, JB, GB, YF, JCG, WH, BH, NJ, WL, PM, MM, MP, BS, QT, TT, JW, SX, XZ contributed to the v2.1 model development and analysis. CZ and TB contributed to the
post-processing, quality control, and publishing of the data to ESGF. JB, BM, and AB contributed to the WMT analysis and text. KS took the lead in writing the manuscript, with significant contributions from AB, LC, JL, AO, and PG. All authors provided critical feedback and helped shape the research, analysis, and manuscript.

*Competing interests.* One of the co-authors is a member of the editorial board of Geoscientific Model Development.

*Acknowledgements.* This work was supported in part by the Biological and Environmental Research program as part of the Earth System
Model Development (ESMD) program area's E3SM project, funded by the US Department of Energy (DOE), Office of Science. This research used a high-performance computing cluster provided by the BER Earth System Modeling program and operated by the Laboratory Computing Resource Center at Argonne National Laboratory. This research also used resources of the National Energy Research Scientific Computing Center, a DOE Office of Science User Facility supported by the Office of Science of the U.S. Department of Energy under Contract No. DE-AC02-05CH11231. This work was also partially supported by the Regional and Global Model Analysis (RGMA) component of the
Earth and Environmental System Modeling Program of the U.S. Department of Energy's Office of Science BER program via National Science Foundation (NSF) IA 1947282 (DE-SC0022070). Los Alamos National Laboratory is operated by Triad National Security, LLC for the U.S. Department of Energy, National Nuclear Security Administration under Contract 89233218CNA000001. Lawrence Livermore National Laboratory is operated by Lawrence Livermore National Security, LLC, for the U.S. Department of Energy, National Nuclear Security Administration under Contract DE-AC52-07NA27344. The Pacific Northwest National Laboratory is operated by Battelle for the
DOE under Contract DE-AC05-76RL01830. The National Center for Atmospheric Research is sponsored by the NSF of the United States of America under Cooperative Agreement No. 1852977. This paper describes objective technical results and analysis. Any subjective views or opinions that might be expressed in the paper do not necessarily represent the views of the U.S. Department of Energy or the United States Government.



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
