# Peer review of "The DOE E3SM Version 2.1: Overview and Assessment of the Impacts of Parameterized Ocean Submesoscales"

_Geoscientific Model Development, 2024_

## Author Comment (AC1)

**Response to the GMD Executive Editor**

We appreciate the Executive Editor bringing our attention to the lack of documentation for the permanent archive of the code base. We have now added the DOIs for both the v2.1 (DOI: 10.5281/zenodo.7527568) and v2 (10.5281/zenodo.5563151) versions of the code used in this manuscript in the Code Availability section of the revised manuscript.

Sincerely, the authors.

---

## Author Comment (AC2)

**Response to Reviewer 2**

We appreciate the constructive and insightful comments from the reviewer. A number of changes have been made to the text at the reviewer's request and we feel that the revised paper is now substantially clearer and stronger as a result. Detailed replies to each of the reviewer's points (in blue italics) are provided below, and changes to the text have been highlighted in red in the revised paper.

*"The DOE E3SM Version 2.1: Overview and Assessment of the Impacts of Parameterized Ocean Submesoscales" has two aspects: it is the report detailing the new version of this model (highly appropriate for GMD) and it is a scientific report on the consequences of including the 2011 version of the MLE parameterization into this model. The latter is not novel in application, but instead compares the sensitivity of the E3SM vs. that of other models incorporating this scheme. The basic result is that upper ocean biases and RMSEs are improved, which then improves many climate metrics. I have only minor suggestions for the authors to consider.*

**Minor Comments:**

**Comment 1:** *Using a constant $L_f$ is not a common choice, although it is not unprecedented. The authors should review the results of extensive testing of different choices of $L_f$ as constants and as variations with mixed layer deformation radius in Calvert et al. (2020).*

We thank the reviewer for bringing our attention to this typo. $L_f$ does indeed vary in our implementation with mixed layer deformation radius, as is suggested in Eq. 13 of Fox-Kemper et al. (2011) and is used in Calvert et al. (2020), and rather $L_{f,min}$ is what is set as a constant. This has now been clarified in the following way in the revised manuscript on lines 65-67:

...Eq. 3 as an estimate of the typical local width of mixed layer fronts, set here in this model configuration as the mixed layer deformation radius. While recent work has been done to improve the representation of $L_f$ (Bodner et al., 2023), we use the original formulation from Fox-Kemper et al. (2011) here in this study...

**Comment 2:** *In Fig. 4, which MLD observations are used? Are they consistent with the definition of MLD used? The new SEANOE MLD product is very nice in its incorporation of much more Argo data than previous versions.*

We thank the reviewer for pointing out this discrepancy. Our MLD observations are from ARGO floats from 2000-2017 using the Holte and Talley (2009) method, while our model MLDs are calculated using the $0.03 \, \mathrm{kg \, m^{-3}}$ density threshold method. We have now swapped out the original MLD observations for the SEANOE MLD observations the reviewer has suggested, which uses the same density threshold method and incorporates more Argo data. Our conclusions in the manuscript based upon this change in observational product do not significantly change, but we have made the following changes in the revised manuscript on lines 100-102 to note the use of this new observational data set:

...from ARGO floats100 and the NCEI-NOASS World Ocean Database (WOD; Boyer et al. (2018)) through the SEANOE data product from 1970-2021 for the mixed layer depth (MLD) (de Boyer Montégut et al., 2004)...

***Comment 3:*** *I recognize that there are many centuries of simulation data reported here, but I wonder the extent to which the MLE parameterization is responsible for these changes as opposed to the many other changes made. Are there any runs available for comparison with the v2.1 that differ only in turning off the MLE parameterization?*

We appreciate the reviewer's question as it brings to our attention that this was not initially clear in the manuscript. The only climate changing code change from the v2 to v2.1 configuration is the addition of the MLE parameterization. Most of the code changes listed in Appendix B are stealth features that are not active in these v2.1 simulations, and the remainder are bug fixes that exhibited no significant climate changing impacts. Thus the authors are assuming that the changes seen in this manuscript are due solely to the addition of the MLE parameterization and any feedbacks the addition of the parameterization induced. We have now added the following text on lines 49-52 of the revised manuscript to help clarify this:

...All other features listed in Appendix B are not active in the v2.1 configuration simulations used in this study, and any bug fixes were shown to have no significant climate changing effects in testing, thus we will assume any changes from v2 to v2.1 are due to the addition of the FK11 MLE parameterization and any feedbacks it may induce in the model...

Sincerely, the authors.

---

## Author Comment (AC3)

**Response to Reviewer 1**

We appreciate the constructive and insightful comments from the reviewer. A number of changes have been made to the text at the reviewer's request and we feel that the revised paper is now substantially clearer and stronger as a result. Detailed replies to each of the reviewer's points (in blue italics) are provided below, and changes to the text have been highlighted in red in the revised paper.

*In the manuscript titled "The DOE E3SM Version 2.1: Overview and Assessment of the Impacts of Parameterized Ocean Submesoscales", the authors develop the version 2.1 of U.S. Department of Energy's Energy Exascale Earth System Model (E3SM) with new updates, by adding Fox-Kemper mixed layer eddy parameterization. The improved eddy parameterization simulates the submesoscale instabilities, especially the mixed-layer eddies slump the fronts. As it is triggered by the mechanism of amplitude of lateral density fronts in the weakly stratified surface mixed layer. After adding the mixed layer eddy parameterization, the simulation for energy exchanges, radiation heat transfer, and adiabatic motion of dry and moist air in the mixed layer are improved.*

**General Comments:**

*This paper is comprehensive and well-supported by evidence, providing an accurate assessment of the effectiveness of mixed layer eddy parameterization in global ocean model of E3SMv2. It presents significant advancements relevant to optimizing the overturning streamfunction. This article is well-written, logically coherent, and structurally sound. Therefore, the following discussion focuses only on specific issues and does not affect the overall scientific validity of the paper. For these reasons, I believe that the manuscript can be accepted for publication by the GMD after minor revision. Below, I have some specific suggestions for the authors.*

**Specific Comments:**

*Comment 1: Line #307: The link of a full list of all code changes made from v2 to v2.1 should be updated, the hyperlink in the PDF does not direct to the correct section. After manually copying the link, there are only two tags, the Atmosphere and Land tags on Github. There is no way to find the corresponding code based on Appendix B, such as River, Sea Ice etc. Reading Fortran code directly is something that often happens when running meteorological models. It is recommended to update the tags or provide an index, then reader could find scripts after reading Appendix B. Additionally, is the repository only open to project developers, or does the entire Github community have permission to modify the code? It seems visitors can leave comments on each line of code.*

We appreciate the reviewer pointing out the issues with the hyperlink and missing tags. We have now updated the link to include the full list of code changes going from the v2 to v2.1 configuration. Filtering these code changes for specific component tags should now reveal the full extent of each change listed in Appendix B along with pointers to the changes made in the fortran code.

Per the reviewer's second point, the repository is open to the public and anyone can make modifications and pull requests, but there is no guarantee as to whether or not the code will be merged in.

*Comment 2: To prevent readers from constantly switching between this paper and Bodner et al., 2023 to reference the key equations in this study, it is recommended to further explain these two variables. The following section is a reference. In equation (1), the physical meaning of the variable $\psi$ is streamfunction, the gradient of the streamfunction can indicate the bolus velocity. $\mu(z)$ vertical fluxes at height z. Equation 2 is vertical structure function, so the vertical fluxes will be vanished when height is below the mixed layer ($z < -H$), and approach zero when z is almost zero.*

We thank the reviewer for this suggestion and have implemented the following changes in the revised manuscript (lines 59-67) that should help the reader understand these variables more readily without referencing previous literature:

...where $C_e$ is an efficiency coefficient, $\Delta_S$ is the local model grid-scale dimension, $H$ is the mixed layer depth, $\bar{b}^z$ is the depth-average buoyancy over the mixed layer, $\hat{\mathbf{z}}$ is the unit vertical vector, $f$ is the Coriolis parameter, $\tau$ is the time needed to mix momentum across the mixed layer, $L_{f,min}$ is a limiting value of $L_f$ to guarantee stability (typically 0.2-5 km), and $N$ is the buoyancy frequency. Eq. 1 can be physically interpreted as an overturning streamfunction that produces a bolus velocity ($u_{MLE} = \nabla \times \Psi$) that acts to slump fronts and provide MLE fluxes to tracers, Eq. 2 as a structure function for the vertical fluxes that has a maximum in the middle of the mixed layer and vanishes to zero at the surface and beneath the mixed layer, and Eq. 3 as an estimate of the typical local width of mixed layer fronts, set here in this model configuration as the mixed layer deformation radius. While recent work has been done to improve the representation of $L_f$ (Bodner et al., 2023), we use the original formulation from Fox-Kemper et al. (2011) here in this study...

*Comment 3: Fig.1, This SSH result is contrary to expectations. Is it possible that there is an issue with reduced deep water formation in the North Atlantic that should feed into the AMOC process (line #134)? Or air-sea interaction? Overlapping lines make it difficult to determine how many lines there are. It is recommended to use different symbols instead and slightly separate the lines in x-axis at overlapping positions to show the location of each point and indicate this in the figure caption.*

Per the reviewer's suggestion, we have remade Fig. 1 to use different symbols for each ensemble member and have spread them out along the x-axis for increased clarity. We have also added in correlations to the plot and an SSH anomaly time series plot (now Fig. 2) to help better understand what is occurring with SSH. The v2 piControl simulation has an initial, almost globally-uniform jump in SSH of 1.5-2cm in the first few years that does not happen in the v2.1 piControl simulations. This global offset persists throughout the piControl and each of the historical ensemble simulations. This leads to a decreased RMSE for v2. Plotting a time series of the global SSH anomalies for the historical period indicates that v2 and v2.1 are very similar, with the ensemble spread overlapping the entire time period, and have trends on par with observations. Additionally, correlations for SSH increase going from the v2 to v2.1 configuration, indicating better model patterns in SSH, and thus better SSH gradients. We have added the following text into the revised manuscript (lines 104-123) and the above noted figure changes to help explain this:

...Correlations serve to evaluate the model's ability to represent spatial patterns, while RMSE

evaluate the model's representation of magnitude (although spatial patterns are also represented in RMSE), in comparison to observations. All correlation quantities increase from the v2 to v2.1 configuration, indicating a better model representation of spatial patterns due to the presence of the MLE parameterization. This increase occurs across all ensemble members, with the most notable increase in the SST correlation, going from 0.725 in v2 to 0.99 in v2.1. For RMSE, although they are relatively modest, SST, SSS, and EKE see a small global bias reduction going from the v2 to v2.1 configuration, and this reduction is seen across all ensemble members. While the mean MLD RMSE for the v2.1 configuration is slightly less than the v2 configuration, the ensemble spreads are essentially overlapping, indicating little difference between the two. Climatological maps of MLD biases discussed later in Figure 5 indicate that the MLDs have changes in regional biases, where some regions see improvements and some degradation going from the v2 to v2.1 configuration, that compensate for each other when looking at this global RMSE metric.

There is an increase in the global SSH RMSE, which can be attributed to an initial ∼1.5-2 cm global increase in the first few years of the v2 *piControl* simulation that is not present in the v2.1 *piControl* simulation. This ∼1.5-2 cm offset remains throughout the v2 *piControl* and each *historical* ensemble simulation. Plotting maps of global *v2/v2.1 historical* climatological SSH (not shown) reveal this step decrease to be globally uniform going from the v2 vs v2.1 configuration, and time series comparing SSH anomalies from v2 and v2.1 shown in Figure 2 show the global anomaly ensemble spreads for the two different model versions to be essentially overlapping throughout the *historical* simulations and having a similar trend over time to observations. Taking this into account and the increased correlation metric for SSH (indicating better model SSH gradients), we believe overall the v2.1 configuration does not exhibit a degraded global climatological SSH in comparison to the v2 configuration. In order to understand regionally where each of the bias changes for MLD, SST, and SSS occur, we next dive into a series of ocean climatological maps...

***Comment 4:*** *Fig.4, if the discussed region, "western boundary current in the North Atlantic around the region of the RAPID array," and "the west transect of the Overturning in the Subpolar North Atlantic Program array" are highlighted with a box in the figure, they would be easier to understand. If the decrease in MLD is caused by a very small increase in the magnitude and extent of the northward limb, then what is the direct cause of the "very small increase"? Is it because the mixed layer eddy parameterization enhanced the bolus velocity of eddies? In E3SM version 2, is the bolus velocity represented by a constant? Could the authors explain this in more detail?*

We thank the reviewer for pointing out this confusing language. It appears that text was describing features in a previous version of the figure that are no longer present in the current version. We have now removed this text from the revised manuscript.

Per the reviewer's second part of the question: in v2 the mesoscale eddy bolus velocity is non-zero and not constant, but the GM coefficient that goes into the calculation of the mesoscale eddy bolus velocity is constant. This mesoscale eddy bolus velocity is the same in the v2.1 configuration. However, there is no submesoscale eddy bolus velocity in v2, that only appears in the v2.1 configuration and it is not constant.

***Comment 5:*** *Fig.6, this is just a suggestion: could the shaded areas in CMIP6 be slightly enhanced, for example by reducing transparency or using borders? The colors are too faint and hard to distinguish. This requires locating the Python script used for plotting in PMP. If it is too difficult, the authors may decide to update it or not.*

Per the reviewer's suggestion, we have darkened the shaded CMIP6 violin areas in both Figs. 6 and 7 (now Figs 7 and 8) for clarity.

***Comment 6:*** *Line #163, what could be the possible reason for the lack of significant changes in large-scale extratropical modes here? In the discussion section (line #301), the reason is also not explained here.*

We acknowledge the importance of the reviewer's question. Although the limited size of samples and the influence of interannual variability limits drawing a robust conclusion, we agree with the reviewer's point that it would be useful to discuss it as a part of the manuscript. In response, we have added the following text to the discussion section on lines 331-336 of the revised manuscript:

...Orbe et al. (2020) distinguished two classes of model improvement: (1) "those that rely on a threshold of model representation that is crossed at a distinct moment in model development", and (2) "improvements that rely on more gradual, collective improvements in processes." They argue that the performance evolution of extratropical modes of variability likely fall into the second category, e.g., due to enhancements in base climate representation and relevant processes, which might be evidenced via mixed influences across different modes and seasons. Additionally, the sample size available for this study limits any robust conclusions regarding performance changes in the simulation of extratropical modes of variability...

***Comment 7:*** *Fig. 14 and 15, it seems that a strong stratification buffer has been enhanced and interior stratification has been eroded in the v.2.1 model compared with v2. Is this solely due to the addition of the mixed layer eddy parameterization?*

Given that the only climate changing code change from the v2 to v2.1 configuration is the addition of the MLE parameterization, the authors have to assume that these changes are attributed to that change, plus any feedbacks the additions of the parameterization induced. Most of the code changes listed in Appendix B are stealth features that are not active in these v2.1 simulations, and the remainder are bug fixes that exhibited no significant climate changing impacts. We have now added the following text on lines 49-52 of the revised manuscript to help clarify this:

...All other features listed in Appendix B are not active in the v2.1 configuration simulations used in this study, and any bug fixes were shown to have no significant climate changing effects in testing, thus we will assume any changes from v2 to v2.1 are due to the addition of the FK11 MLE parameterization and any feedbacks it may induce in the model...

Sincerely, the authors.

---

## Author Comment (AC4)

Dear Editor,

We appreciate your consideration of our manuscript, which has been submitted for publication in *Geoscientific Model Development.* Detailed replies to each of the reviewers' comments and suggestions are outlined in the attached response letters and noted in the marked revised manuscript in red text. We thank the reviewers for their suggestions, and feel that the revised paper is an improvement over the previous version.

Detailed descriptions of the changes made to the text can be found in the replies to the reviewers and in the marked revised manuscript, but the most significant changes are summarized below.

1. Figure 1 has new additional panels showing global MLD, SST, SSS, SSH, and EKE correlation values and the text in Section 3.1 has been extended to discuss these and address Reviewer 1's question about the global RMSE increase in SSH.

2. We have now added a new Figure 2 showing the time series of the global SSH anomaly to further help address Reviewer 1's question about the global RMSE increase in SSH.

3. Additional details describing the MLE parameterization have been added to Section 2.1.

4. The hyperlink for the full list of code changes from v2 to v2.1 has been updated.

5. We have added DOIs for the permanently archived v2 and v2.1 code bases and have updated the Code Availability section to reflex this.

We believe that these changes and the attached letters address all of the comments from the reviewers and the GMD Executive Editor. We thank you for your consideration of our work.

Sincerely, the authors.